# Public Health Policy, Political Ideology, and Public Emotion Related to COVID-19 in the U.S

**DOI:** 10.3390/ijerph20216993

**Published:** 2023-10-29

**Authors:** Jingjing Gao, Gabriela A. Gallegos, Joe F. West

**Affiliations:** 1Department of Management, Policy and Community Health, School of Public Health, The University of Texas Health Science Center at Houston (UTHealth Houston), El Paso, TX 79905, USA; jingjing.gao@uth.tmc.edu; 2College of Health Sciences, The University of North Carolina at Pembroke, Pembroke, NC 28372, USA; joe.west@uncp.edu

**Keywords:** social media, COVID-19, Twitter, public health, public health policy

## Abstract

Social networks, particularly Twitter 9.0 (known as X as of 23 July 2023), have provided an avenue for prompt interactions and sharing public health-related concerns and emotions, especially during the COVID-19 pandemic when in-person communication became less feasible due to stay-at-home policies in the United States (U.S.). The study of public emotions extracted from social network data has garnered increasing attention among scholars due to its significant predictive value for public behaviors and opinions. However, few studies have explored the associations between public health policies, local political ideology, and the spatial-temporal trends of emotions extracted from social networks. This study aims to investigate (1) the spatial-temporal clustering trends (or spillover effects) of negative emotions related to COVID-19; and (2) the association relationships between public health policies such as stay-at-home policies, political ideology, and the negative emotions related to COVID-19. This study employs multiple statistical methods (zero-inflated Poisson (ZIP) regression, random-effects model, and spatial autoregression (SAR) model) to examine relationships at the county level by using the data merged from multiple sources, mainly including Twitter 9.0, Johns Hopkins, and the U.S. Census Bureau. We find that negative emotions related to COVID-19 extracted from Twitter 9.0 exhibit spillover effects, with counties implementing stay-at-home policies or leaning predominantly Democratic showing higher levels of observed negative emotions related to COVID-19. These findings highlight the impact of public health policies and political polarization on spatial-temporal public emotions exhibited in social media. Scholars and policymakers can benefit from understanding how public policies and political ideology impact public emotions to inform and enhance their communication strategies and intervention design during public health crises such as the COVID-19 pandemic.

## 1. Introduction

Infodemiology combines information and epidemiology to analyze geographical patterns of diseases and help inform health policies [1]. It leverages data from various sources, including social network platforms like Twitter 9.0 (X Corp., San Francisco, CA, USA), to understand public reactions including emotions during a public health crisis. Recently, social networks have become an informational instrument in disease management and prevention [2,3,4,5]. Social networks allow the public to express their concerns, experiences, and emotions toward a public health crisis conveniently and promptly on a large platform. Millions of short, geographically localized messages from Twitter 9.0 users are especially valuable in managing public health crises [6,7]. Policymakers could better understand emerging public health crises and issues through information gleaned from social networks [4,8,9,10], as data from some social networks can efficiently capture the spatial-temporal patterns of public health-related emotions, opinions, and behaviors [11].

During the COVID-19 pandemic, social networks, particularly Twitter 9.0, served as a valuable tool to analyze the spatial-temporal patterns of public reactions, especially in the context of a polarized political environment. Scholars have utilized social media data to gauge the collective reactions of the public and study responses to various policy interventions [12,13,14]. Researchers have also studied Twitter 9.0 data to understand disinformation during the pandemic [15] and support for and opposition to reopening policies [16].

Data on emotions retrieved from social networks can provide a snapshot of the public’s reactions to multiple policies and potentially offer some guidelines for use by policymakers [12]. Many studies have explored the public’s sentiment or emotions related to COVID-19 by using Twitter 9.0 data [17,18,19,20]. These studies have analyzed emotions as presented in different countries (such as the U.S., India, Chile, Mexico, Peru, and Spain) and in different languages (such as English and Spanish) [17,18,21]. Twitter 9.0 data have further been used to illustrate the temporal change in public opinions regarding a variety of issues related to COVID-19 [18]. Some scholars have used Twitter 9.0 data to detect the fluctuation in public emotion over time, finding that fear was the dominant emotion during COVID-19 [17,18].

However, despite Twitter 9.0 data’s utility in highlighting the spatial-temporal trends of public emotions during the COVID-19 pandemic [22], many studies overlook the clustering trends (spillover effects) present in information detected through these networks. Clustering is “the propensity for nearby locations to influence each other and to possess similar attributes” [23]. To address this gap, this study examines the clustering trend of COVID-19 emotion patterns on Twitter 9.0. It then analyzes the associations between COVID-19 public health policies, political ideology, and the spatial-temporal trends of public emotions at the county level in the U.S.

### 1.1. Impact of Public Policies and Political Ideology on Emotions

With the tremendous cost in lives, loss of jobs, and the economy’s slowdown, local, state and federal governments adopted various public health policies to curb the spread of the COVID-19 virus, including stay-at-home policies, closures of restaurants, bars and other entertainment-related businesses, bans on large events, and closures of public schools during the early stages of the pandemic. The effectiveness of these policies has been widely studied by scholars [24,25,26,27]. For instance, a stay-at-home policy is more effective in decreasing mobility in Democratic-leaning counties [28]. However, literature on the public’s emotional response under multiple public health policy interventions remains limited in this field. Infodemiology suggests that social media data can reflect emotional patterns; this research further examines whether public health policies affect spatial-temporal patterns of public emotions. This study specifically assesses the impact of stay-at-home policies, restaurant, bar and entertainment-related business closures, bans on significant events, and public school closures on the spatial-temporal patterns of emotions during the early stages of the COVID-19 pandemic.

Moreover, increasing political polarization between Democrats and Republicans makes public policies less efficient, especially in policymaking and implementation [29,30,31]. Political polarization in the U.S. has been a severe issue, as the public and policymakers tend to disregard information that contradicts their political ideology. This leads to a less efficient situation in policymaking and policy implementation [32,33,34]. Political polarization between Democrats and Republicans has provoked mass attitudinal responses to public policies [35,36]. Pre-pandemic studies confirmed that political ideology influences attitudes [37,38], and later studies found that political ideology affects individuals’ health protection actions [28,39]. This study further examines the relationship between political ideology and public emotions toward COVID-19.

### 1.2. Public Policy Responses to COVID-19

California was the first state in the U.S. to announce a stay-at-home policy on 19 March 2020. By 24 March 2020, 14 states had issued stay-at-home policies, and 17 more states (including Alaska and Hawaii) had issued stay-at-home policies by 2 April 2020. Ten states never implemented stay-at-home policies; most are located in the middle of the country. Figure 1a shows the spatial-temporal pattern of stay-at-home policies.

Policies requiring the closure of entertainment facilities and gyms were implemented between 16 March and 3 April 2020. South Dakota was the only state not implementing an entertainment facility and gym closure policy. More information about the spatial-temporal distribution of entertainment facilities and gym closure policies can be found in Figure 1b. Regarding public school closures, all U.S. states had implemented policies closing public schools between 16 March and 3 April 2020. On 16 March 2020, the first day of public school closures, 20 states, including Alaska and Hawaii, issued this policy. Detailed information about the spatial-temporal distribution of public school closures can be found in Figure 1c.

### 1.3. Political Ideology in the U.S.

Figure 2 shows the geographic distribution of the Republican presidential candidate support rate in the 2016 presidential election, indicating that the Republican presidential candidate’s support level was highest in the middle of the country. Though the winning political party changed from 2016 to 2020, the political map reflected similar patterns, with Democratic voters concentrated on the east and west coasts of the country and Republican voters concentrated in the middle. These county election data provide a useful proxy for political leanings in the U.S. in 2020.

### 1.4. U.S. Demographics by County

Social media data have a representative bias [40]. For example, adults with higher educational attainment levels are more likely to use social networks to express their health concerns than their less educated counterparts. This study controls for education at the county level to alleviate bias. Furthermore, many studies using Twitter 9.0 data do not consider contextual information for the tweets, such as the population characteristics and the community’s economic status, so this study also controls for the counties’ population sizes, COVID-19 infection rates, and income.

The corresponding hypotheses are as follows:

Hypothesis H1 (clustering trend): Emotion detected from the COVID-19 Twitter 9.0 has a clustering trend (spillover).

Hypothesis H2 (public health policy): Counties with more days of a stay-at-home policy will experience more negative emotions from COVID-19.

Hypothesis H3 (political ideology): Counties with higher Republican presidential candidate support rates will experience more negative emotions from COVID-19.

## 2. Materials and Methods

### 2.1. Data Sources

This study utilizes composite data from several data sources to test these hypotheses. Twitter 9.0 data were obtained from the School of Data Science at the University of North Carolina at Charlotte from 11 February 2020 to 9 April 2020. Given the importance of geographical location, only geotagged tweets were included. Typically, only 2.02–2.70% of Twitter 9.0 data concerning health topics include GPS information [41]. The Text2emotion (Text2emotion 0.0.5: https://pypi.org/project/text2emotion/ (accessed on 1 May 2021)) Python Package was used to extract emotions from content and detected five emotions (anger, sadness, fear, surprise and happiness) from each tweet. Public health scholars have increasingly used Test2emotion to detect public sentiments [42,43,44]. Data from Johns Hopkins University provided a raw count of deaths and confirmed COVID-19 cases per county, which was available for public use. The 2016 U.S. presidential election results by county provided a proxy measure for political ideology (the data are collected by Tony McGovern from Fox News, Politico, and the New York Times and shared through Github (https://github.com/tonmcg/US_County_Level_Election_Results_08-20 (accessed on 24 October 2023)). U.S. Census Bureau data were used for 2020 county population characteristics, including income and education. This study excludes data from Alaska and Hawaii, as much of the data on these states are unavailable.

Dependent variable. The dependent variable was the aggregated total number of negative emotions per county per week, including anger, sadness, fear, and surprise.

Independent variables. Key independent variables, including the policy’s duration by implementation days for each policy measured starting 9 April 2020, were analyzed at the county level. Political ideology was measured by the Republican presidential candidate’s support or voting rate at the county level in the 2016 Presidential election.

Control variables. The socioeconomic characteristics of counties were controlled to mitigate the potential for representative bias in the Twitter 9.0 data. These control variables were education, income, and population. Education was measured by the percentage of a county’s population with bachelor’s degrees or higher; income was measured by per capita income in each county; the population was measured by the log of the population size by county.

### 2.2. Statistical Methods

This study used Python [45] for data preprocessing, ArcGIS Pro [46] for spatial visualization, and STATA 15.0 [47] for running models. Multiple spatial statistical methods were used for analysis. First, Moran’s I was used to test the spillover effect or the spatial cluster of emotions [48]. Moran’s I tests for spatial autocorrelation and was developed for spatial data. It was selected because one of the goals of this study was to examine the spatial clustering trend of emotions on Twitter 9.0. Second, Zero-Inflated Poisson (ZIP) regression for initial association relationship analysis was used due to the number of dependent variables with zero values. ZIP regression was designed for counting data with excessive zeros [49,50,51,52]. In this study, 82.23% of counties detected zero negative emotions, so ZIP was ideal for the association relationship analysis. The random-effects model reinforced the results by considering the temporal trend of the panel data [53,54]. The negative emotions change temporally, so the random-effects model was used for this study to reinforce the results from ZIP regression analysis. Last, the spatial autoregression (SAR) model was used to analyze the association relationships while considering the spatial components [55,56,57]. Moran’s I test showed spatial dependence of the negative emotion, so SAR was used to reinforce the results from ZIP by considering the spatial dependence. Various spatial-temporal models, such as the fixed-effects model and the random-effects model with spatial weight matrix, are available in multiple software products such as GeoDa 1.22, R 4.3.1, Python 3.12.0, and STATA 15.0 to analyze spatial data [58,59,60]. However, the results from these models are exceedingly complex, and the data size in this study was too large for these models, so this study mainly used the ZIP, random-effects model, and SAR to examine the association relationships.

## 3. Results

### 3.1. Emotions

Table 1 displays the temporal statistics of five emotions (anger, sadness, fear, surprise and happiness) detected in Twitter 9.0 data regarding COVID-19. The numbers indicate the total count of each emotion detected weekly in 2020. The total number of COVID-19 tweets with fear jumped from 165 during the week of 11 February to a peak of 18,023 for the week of 17 March. The number then slowly decreased to 9508 by the week of 9 April. Negative emotions (anger, surprise, sadness, and fear) followed a similar pattern that jumped sharply after the week of 10 March.

Figure 3 shows the daily temporal trends of emotions. It confirms that negative emotions dominated during the COVID-19 pandemic for the study period [61]. Furthermore, Figure 3 shows that fear was the most significant emotion of the five emotions studied, and the first peak of the emotion appeared one day after public school closures and two days before stay-at-home policies were issued. The following analysis aggregated all negative emotions (anger, sadness, fear, and surprise) at the county level as a dependent variable. Figure 4 shows the spatial distributions of all negative emotions. These two figures show that negative emotions studied were primarily distributed on the east and west coasts of the U.S. This phenomenon corresponds with the fact that these areas of the country were hit with outbreaks the earliest during the pandemic in the U.S.

Moran’s I spatial statistical method was applied to test the spatial clustering trend. Table 2 shows the results of Moran’s I test. First, for the week of 11 February, Moran’s I test failed to reject the null hypothesis that the models’ residuals are independent and identically distributed (i.i.d.). This means that the emotion tweets were not statistically spatially clustered together in the early stage of the pandemic in the U.S. Second, however, as time passed, Moran’s I test rejected the i.i.d. hypotheses of emotions for the week of 9 April. This confirms that there were spatial clusters (or spillover effects) of emotions on Twitter 9.0, which supports the visual appraisal of the emotion trend in Figure 4. Furthermore, the change in Moran’s I test results over time indicated that this study should use panel models to analyze the data because time is essential in affecting the spatial distribution of emotions. These two results suggested using the spatial autoregressive model (SAR) as the panel data analysis method.

### 3.2. COVID-19 Policies, Political Ideology, and Public Emotions

A balanced panel sample of 3233 U.S. counties over nine weeks in 2020 from 11 February to 9 April was used in these analyses. Considering the temporal trend of emotions, Table 3 shows the impacts of the total number of COVID-19 infections, the 2016 Republican presidential candidate support rate, and COVID-19 public health policies on the negative emotion count detected in Tweets per week per county. As expected, counties with more total cases of COVID-19 infections tended to have higher negative emotion rates detected from Twitter 9.0, which confirms hypothesis H1. This result is consistent across three models: ZIP, random effects, and SAR. This shows that counties with more COVID-19 infections had a higher negative emotion rate and that Twitter 9.0 data could reflect the public’s emotions based on the spatial-temporal seriousness of the public health crisis.

Regarding stay-at-home policies, the results are consistent across three models. The longer that counties had been impacted by stay-at-home policies, the more negative emotions detected in those counties. These results support hypothesis H2. However, regarding the impact of political ideology on public emotion, the results are not consistent across these models. To better understand the relationships between stay-at-home policies, the Republican presidential candidate support rate, and negative emotions, the interaction effect between the duration of a stay-at-home policy and the Republican presidential candidate support rate on negative emotions was examined. The interaction effect results are shown in Figure 5. Interestingly, as the duration of a stay-at-home policy in each county increased, the predicted number of negative emotions in Democratic-leaning counties with a Republican presidential candidate support level at 10% in the 2016 presidential election increased from 0 to 30 per county per week; however, the predicted number of negative emotions in Republican-leaning counties with a Republican presidential candidate support level at 90% in the 2016 presidential election decreased from 20 to 0. This shows how the public’s emotions changed during the first month of the implementation of stay-at-home policies for counties with different political ideologies.

The random-effects model and SAR model for public school closures show that as the number of days increased, the public’s negative emotions decreased in that county. This indicates that the public’s emotions supported public school closures at the pandemic’s beginning. Based on spatial data analysis results of the panel data, policies like social gathering bans, restaurant closures, and gym closures were not significantly associated with the public’s emotions.

Regarding the socioeconomic status of the counties: (1) the higher per capita income the county had, the more negative emotions the county experienced; and (2) the higher the percentage of people of color in a county, the more negative emotions the county experienced. Interestingly, the negative emotional responses detected likely stemmed from disparate rationales. During the pandemic, people of color were disproportionately negatively impacted in various ways, including mortality [62]. For counties with higher per capita incomes, negative emotions may have been due to restrictions on businesses, sales losses, and economic uncertainty [63].

## 4. Discussion

In this study, we sought to understand public emotional responses by county to public health policies implemented in the early weeks of the COVID-19 pandemic. Other studies have examined state-level public perceptions regarding support or opposition [16]. A better understanding of the public’s emotional response to policies at the county level may offer important insights for policymakers as they consider how to tailor implementation and communication strategies in their areas and the spillover effects of their efforts in surrounding communities.

Negative emotions of the public on social media began to show social dependence or spillover effects by the end of February 2020, confirming that Tobler’s first law applies to the public’s emotions as seen through social media. The spillover effect of public emotions toward COVID-19 means that one county’s public emotions affect the emotions of neighboring counties. Furthermore, the change in spillover effects confirms a study by Dredze et al. [64] concluding that timing matters in analyzing Twitter 9.0 data with geolocation. Our findings are also consistent with a previous study in which fear was found to be the dominant emotion during the pandemic [17]. Counties with more COVID-19 infections showed significantly higher negative emotions, which indicates that the public emotions detected from Twitter 9.0 could reflect the public’s emotional footprint in real time, as the public’s health was being threatened by an emerging health crisis [65,66]. With regard to political polarization and partisanship, the results of this study provide additional evidence of negative emotions consistent with studies indicating that during COVID-19, Democrats “perceive[d] higher risk, place[d] less trust in politicians to handle the pandemic, [were] more trusting of medical experts such as the WHO, and [were] more critical of the government response” [67]. The mistrust of politicians at that time may have driven the negative emotions shown by Democratic voters in the early weeks of the pandemic.

This study provides insight into the U.S. public’s emotional state at the county level in the early weeks of the COVID-19 pandemic. By better understanding how emotions are impacted by policy change, the role of partisanship in emotional response to policy change at the county level, and the geographic scope of the emotional reactions to policy changes, decisionmakers can craft better, more appropriate communication and policy implementation plans. For example, if one county is expected to pass a policy that produces emotions of fear or sadness, its neighboring county might prepare and deploy a communications plan that educates its residents about the scope of that policy and whether and how it might affect them. The messaging may also include reassurances to mitigate the likelihood of spillover effects of the policy passed in the neighboring county. In better tailoring messages and anticipating the emotional reactions of residents, policymakers can more effectively communicate with their constituents and reduce the negative emotions that may follow policy change.

There are several limitations to this study. First, this research cannot confirm causal relationships or predict future results. Second, even though the representation issues of using Twitter 9.0 data were mitigated by controlling for education, population size, and income, Twitter 9.0 data still suffer from technical problems. Social media data were created for direct business purposes, so they are vulnerable to companies’ modification of data collection algorithms to increase profits [68]. Additionally, the availability of only a small proportion (2.5%) of geolocation data limits the strength of our results. Apart from these biases, the accuracy of the Twitter 9.0 surveillance system decreases with “chatter messages”, which are caused by media attention [69].

## 5. Conclusions

Public health policies such as the stay-at-home policy affect the number of negative emotions in a county. As the number of days of a stay-at-home policy increases, the public’s negative emotions increase. This result indicates that future studies regarding the use of social network data for public emotion studies should consider the impact of policy interventions. Furthermore, political ideology affects county-level emotional reactions. Counties with low support rates for the Republican presidential candidate had stronger negative emotions toward COVID-19. This confirms that conflict extension or political polarization impacts public emotional reactions toward a public health crisis. Regarding the impacts of counties’ socioeconomic characteristics on the public’s emotions, this study also found that counties with higher percentages of people of color or counties with higher per capita incomes had higher rates of negative emotions toward COVID-19. Studying the spatial-temporal relationship between policies and the public’s emotional reactions at the time will help policymakers understand the impact of decisions on public sentiment. In summary, this study shows the significance of big data generated through modern communication technologies in enhancing our understanding of the public’s emotional responses to health crises. Additionally, policymakers should be mindful that public policies and political ideology play pivotal roles in shaping public emotions.

## Figures and Tables

**Figure 1 ijerph-20-06993-f001:**
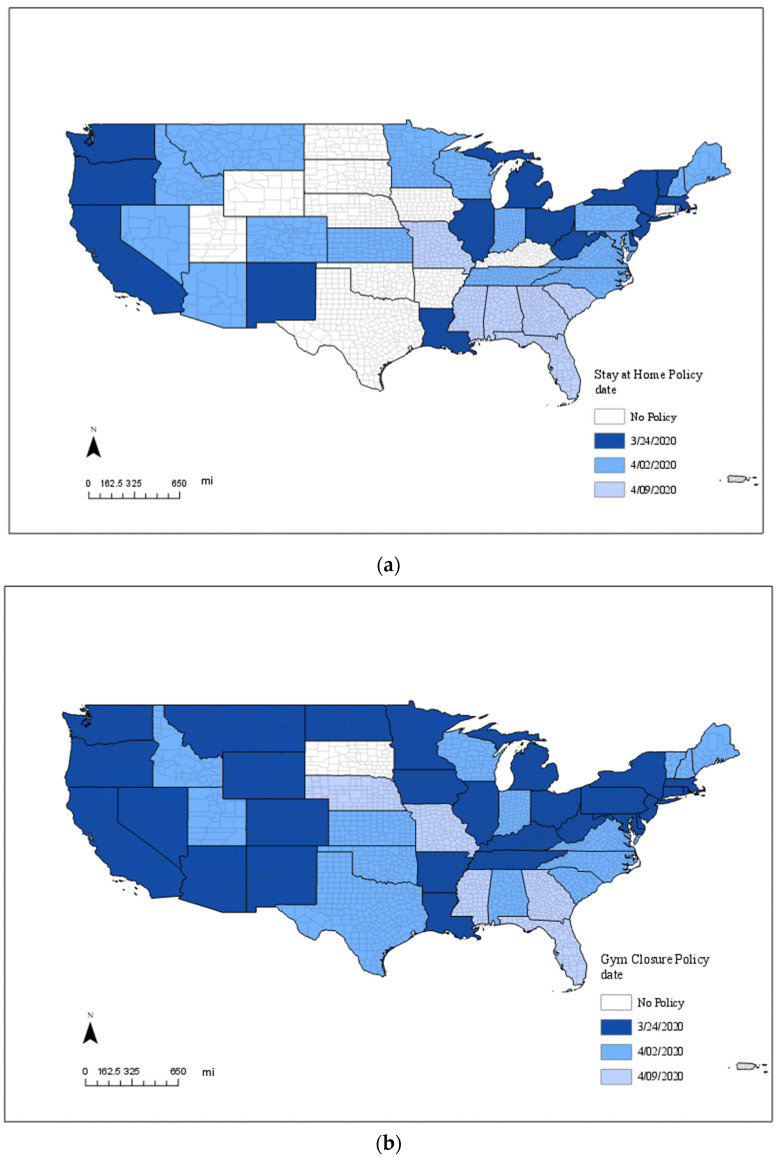
(**a**). Implementation Dates of Stay-at-Home Policies by State. (**b**). Implementation Dates of Entertainment Facility and Gym Closure Policies by State. (**c**). Implementation Dates of Public School Closure Policies by State. Note: White indicates no policy in this state; dark blue indicates policies implemented on or before 24 March 2020; medium blue indicates policies implemented between 25 March and 2 April 2020; and light blue indicates policies implemented between 3 April and 9 April 2020.

**Figure 2 ijerph-20-06993-f002:**
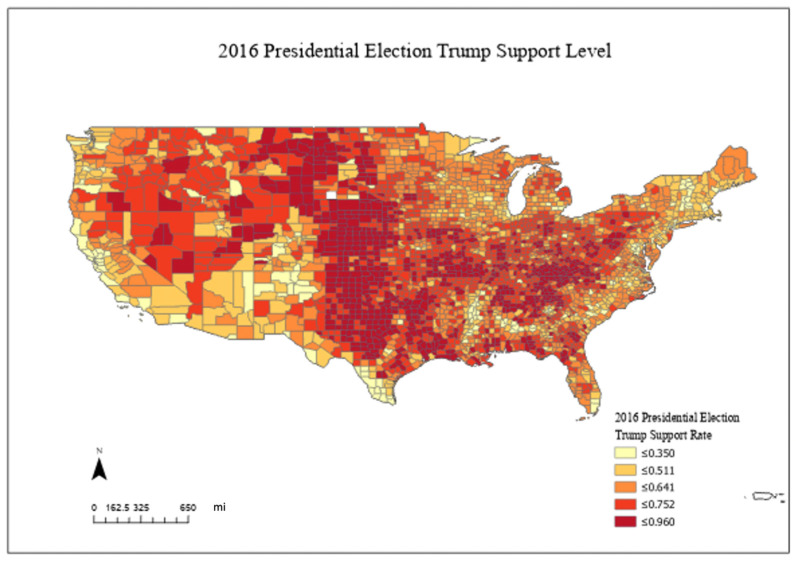
Spatial Distribution of Republican Presidential Candidate Support Rate in 2016 Presidential Election. Note: The intervals are produced by using the method of Natural Breaks (Jenks).

**Figure 3 ijerph-20-06993-f003:**
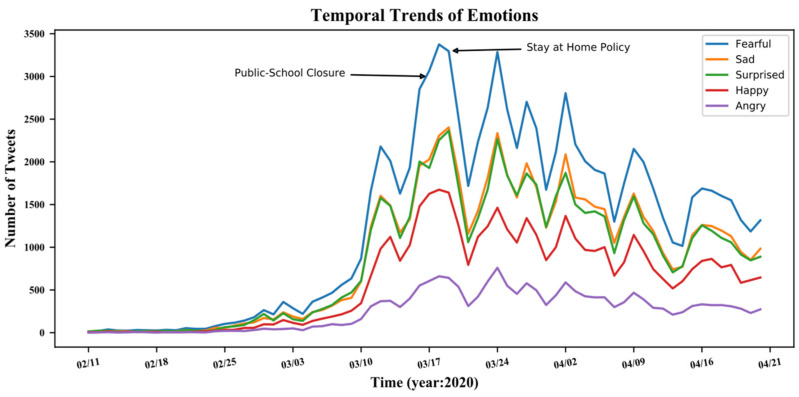
Temporal Trends of Emotions Regarding COVID-19. Note: The lines represent the total number of emotions (anger, happiness, surprise, sadness, and fear) over time by day in the US. The *X*-axis represents time, while the *Y*-axis represents the total number of tweets for each emotion for the week. 02/11 represents 11 February 2020, and 02/18 represents 18 February 2020, etc.

**Figure 4 ijerph-20-06993-f004:**
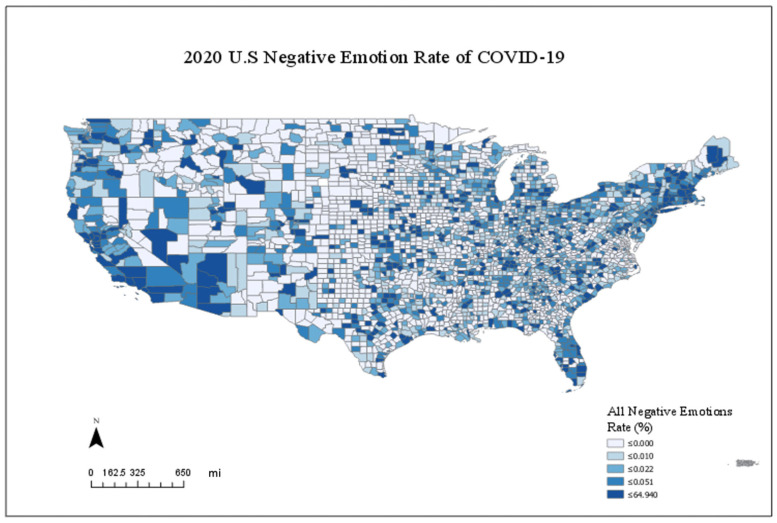
Spatial Distribution of Composite Rate of All Negative Emotions. Note: Negative emotions include anger, surprise, sadness, and fear. The map is normalized by population size.

**Figure 5 ijerph-20-06993-f005:**
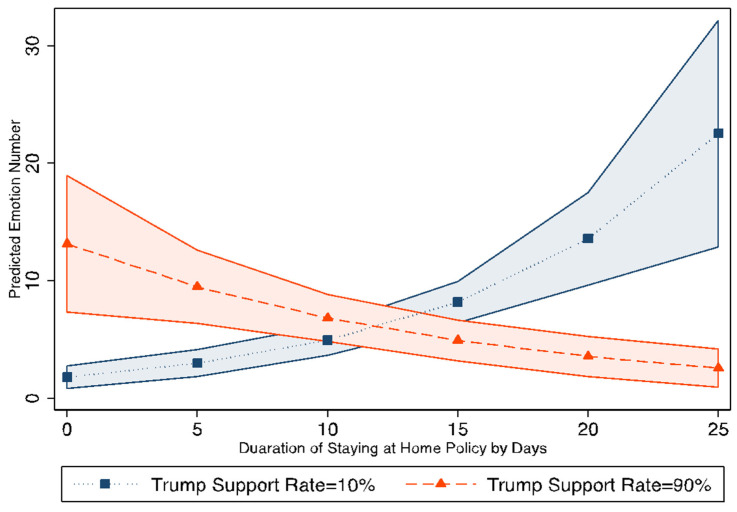
Interaction Effect of Stay-at-Home Policies and Republican Presidential Candidate Support Rate. Note: The orange area and the blue areas are the 95% confidence intervals.

**Table 1 ijerph-20-06993-t001:** Temporal statistics of emotions.

Time (2020)		Happiness	Anger	Surprise	Sadness	Fear
11 February	Sum	49	25	101	102	165
	Max	5	4	19	11	24
18 February	Sum	71	39	143	161	276
	Max	13	8	36	22	55
25 February	Sum	492	214	900	839	1269
	Max	48	21	107	82	125
3 March	Sum	1103	499	1917	1857	2800
	Max	62	33	117	116	169
10 March	Sum	5836	2176	8314	8374	11,685
	Max	367	113	494	471	670
17 March	Sum	8935	3622	11,786	12,419	18,023
	Max	510	238	731	726	1054
24 March	Sum	7796	3474	11,517	11,697	16,232
	Max	463	198	692	733	932
2 April	Sum	6208	2622	8836	9320	12,292
	Max	333	164	479	571	736
9 April	Sum	4709	1889	6714	6934	9508
	Max	303	110	518	502	673
Total	Sum	35,199	14,560	50,228	51,703	72,250
	Max	510	238	731	733	1054

Note: Max indicates the highest number of tweets indicating a particular emotion in one U.S. county for each week studied in 2020.

**Table 2 ijerph-20-06993-t002:** Statistical tests of spatial autocorrelation by Moran’s I.

Time (2020)	Happiness	Anger	Surprise	Sadness	Fear
11 February	3.28	0.10	0.06	0.00	0.02
18 February	8.67 **	0.27	0.03	0.00	2.16
25 February	51.48 ***	46.30 ***	35.96 ***	78.68 ***	81.18 ***
3 March	97.97 ***	49.18 ***	67.71 ***	68.49 ***	87.91 ***
10 March	91.05 ***	91.18 ***	76.32 ***	80.21 ***	79.75 ***
17 March	114.33 ***	83.57 ***	87.00 ***	96.21 ***	105.64 ***
24 March	102.92 ***	109.13 ***	97.51 ***	96.54 ***	98.50 ***
2 April	130.50 ***	80.19 ***	133.22 ***	113.24 ***	120.64 ***
9 April	113.42 ***	103.77 ***	84.57 ***	84.80 ***	106.81 ***

Note: **, and *** indicate significance at 0.01, and 0.001 levels, respectively. The null hypotheses had no spatial autocorrelation.

**Table 3 ijerph-20-06993-t003:** Pooled Ordinary Least-Squares (OLS), fixed-effects, and random-effects models.

Variables	(1)	(1)	(2)	(3)
	ZIP	ZIP	Random Effects	SAR (9 April)
COVID-19 infections	5.24 × 10^5^ ***	−3.59 × 10^5^	0.0107 ***	0.0130 ***
	(1.23 × 10^5^)	(0.000103)	(0.000553)	(0.000911)
2016 Republican presidential candidate support rate	2.432 ***	−0.198	67.82 ***	−7.744
	(0.586)	(0.187)	(9.981)	(9.910)
Per capita income	−4.08 × 10^6^	3.94 × 10^5^ ***	0.000365 *	0.000752 *
	(6.88 × 10^6^)	(5.45 × 10^6^)	(0.000181)	(0.000309)
65 years old population rate	1.217	−0.915	9.003	1.614
	(1.589)	(0.536)	(15.52)	(23.98)
College degree rate	0.0176 **	−0.0473 ***	0.0906	−0.00689
	(0.00589)	(0.00369)	(0.128)	(0.207)
People of color rate	0.786 *	1.228 ***	25.54 ***	20.34 *
	(0.315)	(0.173)	(5.566)	(8.789)
Log population	0.624 ***	−0.993 ***	6.154 ***	9.645 ***
	(0.0589)	(0.0207)	(0.553)	(0.996)
Stay-at-home policy	0.121 ***	−0.00484	4.922 ***	0.736 **
	(0.0214)	(0.00436)	(0.479)	(0.280)
Social gathering ban	−0.00600	0.000866	−0.191	−0.0781
	(0.00810)	(0.00442)	(0.133)	(0.277)
Public-school closures	−0.00704	0.00889	−0.531 **	−1.715 ***
	(0.0128)	(0.00618)	(0.202)	(0.442)
Restaurant closures	−0.0127	−0.0133 *	−0.00194	0.0840
	(0.0117)	(0.00653)	(0.204)	(0.442)
Gym closures	−0.00659	0.0109 **	−0.126	−0.195
	(0.00930)	(0.00420)	(0.144)	(0.247)
Stay-at-home policy × 2016 Republican presidential candidate support rate	−0.208 ***		−7.077 ***	0.0130 ***
	(0.0414)		(0.727)	(0.000911)
Constant	−5.947 ***	12.18 ***	−105.3 ***	−7.744
	(0.925)	(0.336)	(11.38)	(9.910)
Weight (DV: Negative)				−0.493 ***
				(0.117)
Observations	28,017	28,017	28,017	2624
Number of counties			3113	

Standard errors are in parentheses, *** *p* < 0.001, ** *p* < 0.01, * *p* < 0.05.

## Data Availability

The 2016 U.S. presidential election data are collected by Tony McGovern from Fox News, Politico, and the New York Times and shared through Github (https://github.com/tonmcg/US_County_Level_Election_Results (accessed on 24 October 2023)).

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
