# Peer review of "Public Health Policy, Political Ideology, and Public Emotion Related to COVID-19 in the U.S"

_ijerph, 2023, doi:10.3390/ijerph20216993_

Round 1
Reviewer 1 Report
Comments and Suggestions for Authors
An indeed very interesting topic, very well documented and presented.
1. Remember that Twitter has changed its corporate name (it is now X). You must use its current name, or at least make a relevant note at the beginning of your text.
2. This is a research focused in the United States of America. This fact must be explicitely stated in youw abstract, introduction and probably in the title of your paper (I only found it in line 309)
3. In lines 149 and 152, probably a typing error, there is no Test2Emotion1 Python package, maybe you mean Text2Emotion ? And, furthermore, maybe you should move to spaCy for your next paper....
4. In line 155 digit 2 should be raised (footnote).
5. In Table 1a, all rows denoting min could be erased (all 0's). Furtermore, what do min's and max's denote?
6. Lines 190-192: Happiness on Covid-19?
7. Although nothing can be done, a percentage of 2.5% of the geolocation data availability, sadly induces a very important bias in all such studies. This reviewer has strugled many times with this fact.
8. You might want to check https://doi.org/10.3390/su13116150
Reviewer 2 Report
Comments and Suggestions for Authors
This study explores the spatial-temporal clustering trends of negative emotions associated with COVID-19 and examines the relationships between public health policies, political ideology, and the negative emotions linked to COVID-19.
The research employs various statistical methods, including Zero-Inflated Poisson (ZIP) regression, random-effects models, and spatial autoregression (SAR) with data from multiple sources, such as Twitter, Johns Hopkins, and the U.S. Census Bureau. The findings show that negative emotions related to COVID-19, as derived from Twitter data, demonstrate spillover effects. There are also many other findings discussed in the paper.
The paper is well-written and engaging. Comments are:
1) Please provide citations for all the software used and statistical methods. 2) Discuss some related work in the literature.
3) Provide a detailed discussion of the statistical methods and why you chose them.
